# Algorithm for Autonomous Management of a Poultry Farm by a Cyber-Physical System

**DOI:** 10.3390/ani13203252

**Published:** 2023-10-18

**Authors:** Nayden Chivarov, Kristiyan Dimitrov, Stefan Chivarov

**Affiliations:** Institute of Information and Communication Technologies, Bulgarian Academy of Sciences, 1000 Sofia, Bulgaria; nchivarov@gmail.com (N.C.); schivarov@gmail.com (S.C.)

**Keywords:** cyber-physical systems, automation algorithms, farm automation, smart farming, OpenHAB

## Abstract

**Simple Summary:**

The article presents a Cyber-Physical System (CPS) implemented using open-source software, used to autonomously manage the microclimate in a broiler meat production poultry farm. The system is controlled by a small Raspberry Pi computer. The CPS is built up of multiple sensors, actuators and end devices connected to each other in a network. It provides autonomous control of parameters such as temperature, relative humidity and speed of air flow in the building to ensure optimal conditions for the broiler growth. On the one hand, the system is built with cost-oriented components, but on the other hand, it provides fully autonomous control, without human intervention in all ventilation modes. This eliminates human errors and increases productivity.

**Abstract:**

The article presents a Cyber-Physical System (CPS) for intelligent management of a poultry farm for broiler meat production, with a fully autonomous microclimate control. Innovative concepts have been introduced for automated management and changing parameters according to pre-set conditions and schedules, with the possibility that the parameters of the algorithm can be further adjusted by the operator. The proposed CPS provides for high productivity with minimal production waste, at optimized costs and with minimization of human errors. The CPS is built on the basis of cost-oriented components. A Raspberry Pi 4 8 GB is used as the server, and the free open-source software OpenHAB 3.0 is used to optimize the cost of building the system as much as possible.

## 1. Introduction

The consumption of meat in Europe is gradually increasing, in particular the consumption of chicken meat. In some countries consumption has grown by 30% over the last 10 years, 20% in the case of Bulgaria according to EUROSTAT data [1]. As a result, poultry farms using ever higher levels of automation started being constructed.

Raising chickens requires more environmental control than raising other meat-producing animals such as cows or sheep, for example. Any deviation from the optimal parameters increases mortality, especially in the first days of brooding, and subsequently reduces productivity [2]. On the other hand, broiler breeding is based on a number of sub-sciences, the most important of which are animal breeding, nutrition and reproduction [3].

When raising broilers, the control system needs to monitor parameters such as temperature, relative humidity RH, speed of air flow SAF, differential pressure ∆p and concentrations of harmful gases such as ammonia NH_3_ [4], carbon dioxide CO_2_, carbon monoxide CO and hydrogen sulfide H_2_S [2,5]. Dust and microbial contamination should be monitored as well [2]. The intensity and duration of lighting in the premises and the noise level are important factors [6,7]. It is also crucial to note that the most important parameter that determines chickens’ comfort zone is the felt temperature [2,8,9,10]. It is often different from the air temperature in the premises and depends on many factors: air temperature, floor room temperature, relative humidity RH, speed of air flow and chicken age and plumage. When the air temperature, humidity and wind speed are within acceptable limits, the felt temperature matches the temperature in the premises. At a higher temperature level, in order not to leave the comfort zone, the speed of air flow must be increased [2]. With the proposed methods, the felt temperature can be calculated and used for system operation.

A Raspberry Pi 4 [11], a cost-effective single-board computer, was used as the CPS hardware server, The server uses a Linux-based operating system and is running an OpenHAB [12] open-source server instance. Using open-source server software as the basis of the CPS saves the farmer from spending money on expensive software products. In order for the CPS to provide fully autonomous microclimate control, all the basic parameters of the environment must be monitored with sensors, and the building must also have the required microclimate control systems. For the geographical latitudes of Bulgaria, these are heating, lighting, fresh air fans, circulation fans, automatic fresh air blinds, tunnel ventilation inlets and evaporative cooling pads. Evaporative cooling pads have prevailed over in-house foggers, because they are easier to manage, more efficient and do not wet the litter, which is a cause of harmful vapors formation. A large number of the existing poultry farms in Bulgaria have this equipment in place, but most of them do not have systems for automatic management of all of their equipment. Automation is partial, and highly qualified staff are permanently monitoring the conditions in order to take action and maintain a chicken-friendly environment. The proposed CPS is suitable for such existing poultry farms as well as for newly constructed ones.

## 2. Specifics of Raising Broilers

To maintain a favorable microclimate, the monitored parameters must be within certain limits. The concentrations of harmful gases according to the regulations [5] must be for ammonia NH_3_ < 15 ppm, for carbon dioxide CO_2_ < 3000 ppm, for carbon monoxide CO < 4.4 ppm and hydrogen sulfide H_2_S may not be present. The presence of fine dust particles when grown on litter should be <5 mg/m^2^ and without litter < 3.5 mg/m^2^. Optimum temperature and humidity in the room must be maintained according to the age of the chickens, and fresh air must be provided. Almost 4000 kilotons of ammonia are released annually from the agricultural sector in the European Union. That is about 94% of total ammonia emissions [13].

Wood shavings absorb moisture and thus reduce the release of NH_3_ [14,15]. Aluminum sulfate added to poultry litter also reduces NH_3_ volatilization and reduces chick mortality in the first 3–4 weeks [16].

The internal body temperature of birds is between 39 and 42.2 °C [17,18,19]. In order to achieve maximum productivity, a temperature zone narrower than the comfort zone of the chickens must be maintained. When the temperature zone is properly maintained, the energy surplus from feeding is used for growth and weight gain, rather than for providing temperature comfort to the broiler. This zone is defined by the felt temperature. Therefore, it is necessary to provide optimal conditions for broilers where air temperature, relative humidity and speed of air flow are within fixed limits. Maintaining these conditions ensures not only good performance of broilers, but also ensures that the health and welfare of the birds are not compromised. If the air temperature rises above the fixed limits, a cooling air flow must be applied so that the felt temperature remains unchanged. Some authors [20] recommend a temperature of 30–32 °C for the first week and a decrease with 3–4 °C per week until the chicks reach 4 weeks of age. But the optimum temperature and relative humidity RH to be maintained can be seen in Table 1 [5].

At high ambient temperature, broilers maintain a thermo-neutral temperature by losing heat mainly through conduction, convection, radiation and evaporative cooling [21,22]. Birds do not sweat and cannot cool down efficiently in high temperatures. Relative humidity RH mainly affects breathing. At lower humidity, more water evaporates with the exhaled air, which helps cooling. When moving out of their comfort zone, as the temperature rises, chickens start breathing frequently to increase their cooling. At high RH, evaporation is less and cooling is also reduced. Humidity must be controlled because when it is too high it provides a habitat for microorganisms, thus exposing the birds to the threat of disease [23]. Another sign that broilers are hot is retracting the feathering to the body and continuous raising or lowering of the wings in order to uncover the body from the thick plumage and get better cooling. They also move to locations where the speed of air flow is higher. When they are cold, they gather in groups, crouch and press themselves against the floor. They bristle to increase the insulation of their feathers. When the temperature is in the comfort zone, the SAF should be between 0.2 to 0.3 m/s [2]. When the air temperature rises above the comfort zone, the speed of air flow SAF should increase proportionally to the temperature, to 0.8–1 m/s [5] in younger broilers and to 2.5 m/s or 3 m/s [2] in adults.

For the system to work correctly, the sensors for indoor temperature ti, indoor relative humidity RH, speed of air flow SAF, ammonia NH_3_, carbon dioxide CO_2_, carbon monoxide CO, hydrogen sulfide H_2_S and fine dust particles must be at the level of the chickens. Exceptions are the sensors for measuring the temperature below the roof of the building tih, as well as those for outdoor temperature to, outdoor relative humidity RHout and differential pressure ∆p.

To simplify the algorithm, fine dust particles and microbial contamination sensors are not included in the considered cyber-physical system, because automated actions for their reduction cannot be implemented. The use of these sensors may be for informational purposes only. Sensors can be added, and the system will send a message when the measured values exceed the maximum set.

Birds generate heat and water vapor, which strongly affect the microclimate in the building. They are directly proportional to their weight. The heat released is on average 11.6 kJ/h/kg [2]. So, in a medium-sized building with a capacity of 20,000 broilers with an average weight of 2 kg per broiler, the released heat is 130 kW/h. The same flock can evaporate up to 4000 L of water per day. In addition, birds also exhale carbon dioxide CO_2_, which must be removed with the help of ventilation. Wet litter causes activity of coccidiosis and cholestridia, and leads to weight loss in birds, which reduces productivity [24,25].

Indoor lighting period and light intensity also have an impact on farm productivity. In the first weeks of raising chickens, it does not affect the health status of the chickens, but mainly affects the time of activity and feeding [6,7]. For larger chickens, longer lighting periods and higher light intensities can have the opposite effect. It can lead to cannibalism, fatigue of the chickens, various diseases and even death [26,27,28]. Some authors recommend 16 h of light and 8 h of darkness [29,30], while others report higher productivity with alternating shorter light and dark periods [31,32,33].

Before the chickens are housed, a constant temperature must be ensured for at least 24 to 48 h depending on the type of heating, so that the litter can warm up to 30 °C. If the temperature is too low or high, the mortality rate for the first 7 days can increase from 0.7% to 14% [2].

## 3. Types of Ventilation in a Poultry Farm

In order to achieve a good microclimate in the premises, the ventilation system must be well-designed. It should contain a certain number of ventilators which suck the air from the interior of the building, whereby fresh air enters through multiple blinds with adjustable openings or through large inlets at one end of the building.

Three types of ventilation can be distinguished: minimum ventilation, transitional ventilation or tunnel ventilation.

### 3.1. Minimum Ventilation

Minimum ventilation is applied in cold weather when the heat generated by the broilers is not sufficient to achieve the desired temperature in the building. It must provide fresh air sufficient to maintain the desired parameters and air quality, while at the same time it is not allowed to exceed the norms of harmful gases such as NH_3_, CO_2_, CO and H_2_S [2,5,10]. Attention should be paid to the possibility of frostbite in young chickens, which can greatly increase mortality [2]. But on the other hand, a balance must be achieved between incoming fresh air and energy efficiency when using heating. Minimum ventilation is characterized by the fans operating for a period of time during which they introduce fresh air through multiple ventilation blinds, followed by a waiting period with ventilation stopped and the blinds closed. Over time, it has been found that the most appropriate intervals are about 5 min, with about 30 s of ventilation for each [2]. Also, circulation ventilation should be applied, which stirs the warm air from the top of the building with the colder air lying low down, during the time the ventilation is off. Circulation fans are also called stir fans. Thus, heating savings are achieved which can reach up to 20% in new and well-designed buildings and up to 40% in buildings with conventional heating and high roofs [2]. In addition, circulation ventilation helps to dry the litter, which emits much less harmful gases and provides the necessary air movement of 0.2 to 0.3 m/s. To prevent the incoming cold air from falling directly on the floor and on the chickens, it must enter the building at a certain speed. This results in a mixing of the cold and warm air and a more even temperature. It also prevents the litter from getting wet when it comes in contact with cold air masses. The maintenance of this speed is realized by measuring the differential pressure, which is the difference between the pressure in the building and the external atmospheric pressure. The extractor fans lower the pressure in the building. The blinds are open and the opening should not be too small, so that the fresh air enters evenly throughout the building. The minimum ventilation should operate at a differential pressure ∆p between 17.5 Pa and 30 Pa for good efficiency. This is achieved by adjusting the opening of the blinds and the power of the fans, on condition that the ventilation must provide the required volume of fresh air for broilers.

### 3.2. Transitional Ventilation

Transitional ventilation is used when the heat released by the chickens must be removed from the building to achieve temperature balance. It is similar to minimum ventilation, but in this case the ventilation runs continuously. The blinds are placed in different positions to bring a larger volume of cold air into the building. Here again we have to keep the differential pressure ∆p within the specified limits.

With minimum ventilation and transitional ventilation, the speed of air flow should be maintained within the limits of 0.2 to 0.3 m/s, which does not affect the felt temperature [2].

### 3.3. Tunnel Ventilation

Tunnel ventilation is applied when transitional ventilation is not sufficient to achieve the necessary cooling of the broilers. In this type of ventilation, fans are located on the short side of the building, and large openings called tunnel inlets open at the other end. When the fans are switched on, air enters through the tunnel inlets, creating an even air flow along the entire length of the building. Thus, even at higher air temperatures, the air flow cools the broilers and the temperature they feel is lower than the real temperature in the building. Evaporative cooling pads are installed on the inlets, which help with cooling at high temperatures. Cooling with foggers inside the building is not recommended in a climate similar to that in Bulgaria, because very often wetting of the litter occurs and thus the release of harmful gases and problems with the health of the chickens. This type of cooling is only suitable for very dry climates. This problem does not exist when using evaporative cooling pads. With them, it is important to correctly choose their area so that enough air can enter through the tunnel inlets. The velocity of the air that passes through the pads at the required maximum value of the speed of air flow in the building must not exceed certain values depending on the specifications of the pads. Otherwise, their efficiency decreases.

In these buildings, for all types of ventilation, good sealing of all doors, ventilation blinds and tunnel inlets should be ensured when they are closed. Regular cleaning should be ensured. This ensures that the air will pass exactly where it needs to for maximum efficiency.

## 4. CPS Algorithm

In the proposed autonomous control algorithm in tunnel ventilation mode, the system controls the felt temperature tf, instead of the measured one. The system calculates the felt temperature according to the preset dependencies. It depends on the room air temperature ti, the speed of air flow SAF, the age of the broilers ts1 and the relative humidity RH. Table 2 applies to 7-week-old broilers and shows by how many degrees the felt temperature tf is lower than the air temperature for ti values from 18 °C to 38 °C, and wind values from 0.75 m/s to 2.5 m/s.

An assumption that wind speed SAF ≤ 0.5 m/s does not affect the felt temperature is used in the CPS algorithm. Similar tables have been developed for the operation of the system for different ages of broilers. Some of the data for these tables are taken from various sources [8,9,10,34], and the tables are supplemented by interpolation and extrapolation.

At values of relative humidity RH higher than RHmax for the corresponding age of broilers, the felt temperature tf increases with values indicated in Table 3. It is based on data derived from the dependence of temperature and relative humidity RH when determining the temperature-humidity index THI. Tests carried out with hens [35] show that when maintaining the same THI value, with changing temperature and relative humidity, there is no change in their physical condition and feeding style.

Table 4 shows the parameter notations used in the algorithm presented below.

The felt temperature can be found using the following formula:tf = ti − tcw + tcRH

### 4.1. Building Preparation Algorithm

Figure 1 shows an algorithm for preparing the building before introducing the chickens. This mode should be started at least 24 or 48 h before the chickens are housed. A minimum and maximum maintenance temperature must be set at start-up. The system starts by ventilating the room to provide fresh air and remove smells from the preparatory disinfection.

### 4.2. Main Part of the Work Algorithm

The main working algorithm of the proposed cyber-physical system can be divided into five parts for easier presentation. In the first part (Figure 2), the temperature in the building at the level of the broilers ti is measured and compared with the temperature set by the system. That temperature changes according to the age of the chickens by the timer ts1. It depends on what type of ventilation will be undertaken. The system goes through one cycle of operation, then goes through the ti check again. This is repeated continuously, with one exception in the tunnel ventilation Tu.

The minimum ventilation branches M1 and M2 are almost identical, so we will only look at branch M2 (Figure 3) and note the differences.

#### 4.2.1. Branch of the Minimum Ventilation Algorithm

In the minimum ventilation mode M2, the system first checks for the presence of harmful gases. If a high concentration is not detected, it proceeds to the following actions. The tunnel inlets are closed and the tunnel fans are turned off. Evaporative cooling is also turned off. The heating is switched on, and the power or the number of switched-on heating devices depends on the difference between the optimal temperature tiopt for the current age of the broilers and the measured ti. The blinds are opened in position X, which is determined by the difference between the internal and external temperature ∆t1, and by the age ts1 and the number N of chickens. Age and number determine the live weight of the chickens, which determines what volume of fresh air should be provided.

Position X means that a certain number or all of the blinds can be opened. This is determined by the specifics of the building. The fans for minimum and transitional ventilation, which we will call “ventilation” (different from tunnel ventilation and circulation ventilation), are activated. The number of operating fans and their power is matched to the position of the blinds X to achieve the specified differential pressure ∆p. Based on the age of the chicks, the system sets the ventilation cycle time ts2 to a value between 5 and 7 min. The extension to 7 min is to save energy in the brooding period, when higher temperature and higher relative humidity must be maintained. The system waits for the time ts4 to expire (it is in the order of a few seconds). This time is needed to open the blinds and rotate the fans until a constant differential pressure ∆p is achieved. Due to contamination of the blinds or ventilation outlets of the fans, ∆p may not be within the permissible values and unwanted air movement may occur. Therefore, an additional adjustment of the ventilation power is applied until ∆p reaches the required values. After that, the ventilation continues until timer ts5 expires. The total operation time of the ventilation is the time ts3 which is 30 s. After the ventilation stops, a comparison is made between the indoor temperature at broiler level ti and indoor temperature below the roof tih. If the difference ∆t3 is bigger than the maximum allowed, circulation fans are switched on. After timer ts6 expires, the time of which is the time of one cycle (from 5 to 7 min) minus the time of operation of the ventilation, the circulation fans are switched off if they were switched on. The relative humidity inside the building is then checked. If it is lower than the minimum, the system sends a one-time message with the measured value. To adjust the humidity, the system decreases timer ts3 for the next ventilation cycle. If RH is larger than the maximum value, the system also sends a message, but the time ts3 is increased for the next cycle. When the relative humidity RH is normal, no change is made and it goes directly back to the first part of algorithm and the measurement of ti.

If, while checking for the level of harmful gases, the level of any of them has reached 80% of the maximum permissible value, the system performs the same actions as when they are normal, but the blinds open one position step more at position X + 1. Accordingly, the ventilation changes its power.

If, despite the measures taken to reduce harmful gases in the next cycle, it turns out that they exceed the permissible norms, the system sends a one-time message. This is emergency mode and timer ts2 takes a value of 3 min. The heating is on, but the blinds are opened to maximum value. The ventilation also operates at the corresponding power during the whole period of the ts2 cycle. If necessary, the ventilation power is adjusted. After the ts5 timer expires, a check of the relative humidity is made, and if it is not within the norms a message is sent. No regulation measures are taken, because the priority of actions to remove harmful gases is higher. Then we return to ti measurement mode again.

The differences between minimum ventilation branches M1 and M2 are as follows. When entering the mode, the system sends a one-time message with the measured temperature. The heating always works at maximum power. If the concentration of harmful gases is above the norm, the blinds do not open to the maximum position. They open at position X + 2 to prevent the broilers from getting cold. Everything else is completely identical.

#### 4.2.2. Branch of the Transitional Ventilation Algorithm

In transitional ventilation (Figure 4), the system again first checks for harmful gases. If they are normal, the following actions are applied. Heating, tunnel ventilation and evaporative cooling are switched off, and tunnel ventilation inlets are closed. The blinds are opened in position X and the ventilation is switched on at the corresponding power Y, as in minimum ventilation. The duration of the ts5 cycle is 5 min. Ventilation continues throughout the whole cycle. After the blinds’ positioning time has expired, the differential pressure ∆p is checked, and at the end of the cycle, the relative humidity RH is checked. If the RH is not within the norms, the system sends one-time messages with the measured value, but it does not take any action to regulate it. In this case, supporting the temperature has a higher priority. The cool air entering from the transitional ventilation provides constant movement of the air inside the building. Therefore, it is not necessary to turn on the circulation fans. After the RH check, the algorithm returns to the beginning to measure the temperature ti.

If, while checking for the level of harmful gases, the level of any of them has reached 80% of the maximum permissible value, the system performs the same actions as when they are normal, but the blinds open one position step more at position X + 1. Accordingly, the ventilation changes its power.

If the values are above the permissible ones, emergency mode is entered and the system sends a one-time message. The cycle time is shortened to 3 min to avoid overcooling the broilers and the blinds are opened to their maximum position.

#### 4.2.3. Branch of the Tunnel Ventilation Algorithm

In tunnel ventilation, Figure 5, the system again checks for the presence of harmful gases above the norm first, and this would only happen if there is a problem with the ventilation or inlets, because the volume of fresh air that enters the building is many times greater than that of the transitional ventilation. The system then calculates the felt temperature tf according to formula 1 at the maximum value of wind speed SAF for the corresponding broiler age ts1.

If the calculated temperature tf is lower than the minimum temperature tmin for the broiler age ts1, the system performs the following actions. The optimal wind speed Zopt is calculated. At this wind speed, the felt temperature tf is equal to the optimal temperature topt for the corresponding broiler age ts1. Cycle duration ts2 is set to 5 min. Heating, ventilation and evaporative cooling are turned off. The blinds are closed. The tunnel inlets are opened, and the tunnel ventilation is switched on at the power Y which corresponds to the calculated air speed Zopt. The system waits for timer ts4, which is necessary to open the tunnel inlets and spin the tunnel fans until a constant wind speed is established. The actual wind speed SAF is measured, and if it differs by more than 10% from the set wind speed, the system changes the power of the fans to adjust the wind speed. This can happen if the tunnel inlets or the evaporative cooling pads through which the air passes are contaminated, even when the evaporative cooling is not switched on. After the timer ts5 expires, the system again returns to the mode of measuring the internal temperature ti.

If the calculated temperature tf is between the minimum temperature tmin and the maximum temperature tmax for the broilers age ts1, the system performs the same actions as in the previous branch, but here the power of the fans is set to achieve the maximum value of wind speed SAF corresponding to the broiler age ts1. After the end of the cycle, it goes back to ti check mode.

If the calculated temperature tf is higher than the maximum temperature tmax for the broiler age ts1, the system checks the relative humidity outside RHout. If the measured relative humidity RH is greater than 80%, the system does not turn on the evaporative cooling. It returns to the previous branch where the wind speed is at the maximum according to ts1.

If RHout is less than 80%, evaporative cooling is turned on. The tunnel inlets are opened, and tunnel ventilation is switched on at the power Y which corresponds to the calculated air speed Zmax. Zmax corresponds to the maximum value of the wind for the corresponding broiler age ts1. The system sets the cycle time ts2—5 min (or longer according to the specifics of the evaporative cooling pads). If necessary, after the timer ts4, the power of the tunnel fans is adjusted. By the end of ts2, the cooling pads are moistened, and the temperature ti starts to decrease.

The system recalculates the felt temperature tf. If tf is lower than tmin, the system enters an evaporative cooling stop cycle. First, the system sets a new value of the power of the tunnel fans Y, to establish Zopt. Zopt corresponds to a wind speed at which the felt temperature tf is optimal. SAF is measured and corrections are made if necessary. Evaporative cooling is turned off and the cycle continues until the cooling pads drying time ts drying expires and the system returns to checking ti again.

If the felt temperature tf is between tmin and tmax or higher than tmax, then the system enters a new closed cycle that does not go back to the beginning to check ti. Only the felt temperature tf is checked in this cycle. In both cases, the system first checks the wind speed SAF and adjusts it, if necessary, then checks for harmful gases and for high relative humidity outside RHout. In the first case, the wind speed is maintained such that tf is optimal. In the second case, when tf is higher than tmax, the wind speed is maintained at a maximum value according to age of the broilers ts1, but the temperature is still higher than the maximum. However, this mode is not considered an emergency mode because there is no way to reduce the temperature further. If RHout is above 80%, the system exits the closed cycle and begins to perform the evaporative cooling shutdown procedure. Then returns to the beginning to check ti. If RHout is normal, the system checks tf until tf is detected to be less than tmin, and then initiates the evaporative cooling shutdown procedure.

## 5. Test and Result

The best test scenario is for the system to be fully implemented in a real building with all of the sensors for temperature, humidity and differential pressure described and at least one type of sensor for harmful gases. The ideal test system should also include all of the executive systems described above. Due to the impossibility of creating this experimental setup, the system testing was carried out in two stages. In the first stage, all sensors were connected to the system and tested by reading and recording data. The following sensors were used: DHT22 sensors for temperature and humidity, a SCD30 sensor for CO_2_, the Modbus version of the A2G-50 sensor for differential pressure and a QVM 62.1 for wind speed.

In the second stage, instead of real data from the sensors, synthesized data for temperature, humidity and pressure were given as input to the system via the MQTT Broker. This allowed different conditions to be simulated to check system operation.

During the test, the following initial conditions were set: ts1—14 days, maximum number of broilers. Then the system determined tmin—24 °C, tmax—27 °C, topt—25.5 °C. When the system was started, data were sent to it every 30 s. Table 5 shows only a part of the lines where some of the data have changed. Figure 6 shows the response of the control systems based on the input data received.

The system started as follows. It opened the connected blind to position 2 of a total of seven positions, started the ventilation fan at the set speed corresponding to position 2 of the blind and turned on the heater at 100%. After 30 s, the system turned off the ventilation fan, closed the blind and turned on the circulation fan.

At the 5th minute, the cycle ended and the algorithm started a new one, and the system measured temperature ti and took the following actions: it turned off the circulation fan, turned on the ventilation fan and opened the blind to position 3. The heating remained on full power. After 30 s, the blind was closed and the ventilation fan stopped.

At the 10th minute, the system measured temperature ti and switched from Minimum ventilation mode to Transitional ventilation mode, because temperature ti was higher than topt. The heater was turned off, the blind was opened to position 4 and the ventilation fan was turned on at the appropriate power.

At the 15th minute, the system measured temperature ti and switched from Transitional ventilation mode to Minimum ventilation mode. The heater turned on with PWM modulation at 10%, the blind moved from position 4 to position 3, and the ventilation fan switched to the appropriate power. After 30 s, the ventilation fan stopped, the blind closed and the circulation fan turned on again.

At the 20th minute, the system measured temperature ti and the level of carbon dioxide CO_2_ at above 80% of the permissible value. Therefore, the blind opened to position 4 instead 3, the ventilation fan was turned on, the heater remained at 10% and the circulation fan was turned off. After 30 s, the ventilation fan was turned off and the blind closed.

At the 25th minute, the system measured temperature ti and the level of carbon dioxide CO_2_ above the permissible norm, so it entered emergency mode for 3 min. It sent a message “Warning: High CO_2_ level—3000 ppm”. The blind opened to position 4, the ventilation fan turned on at the appropriate power and the heater increased its power to 50%.

At the 28th minute, the system measured temperature ti and CO_2_ was already below the permissible norm. The blind opened to position 3, the ventilation fan switched to the appropriate power and the heater switched to 10%.

The test above has shown that the system (the algorithm) can successfully switch to different ventilation modes as well as entering and exiting emergency mode. Figure 7 shows how the sensors should be located in a real poultry farm.

In a real situation, the opening of the blinds in any position at different temperatures it must be in accordance with the amount of fresh air that is necessary for the good health of the broilers. Sample values are set for the experimental stage so that the system can be tested.

## 6. Conclusions

Microclimate control systems currently in use in poultry farms cannot switch from one ventilation mode to another by themselves. They maintain the necessary parameters in a fixed ventilation mode, which has to be chosen by the farm staff. The proposed algorithm provides fully autonomous management of the microclimate in a poultry farm. It controls the required temperature, speed of air flow and relative humidity. In any situation, if necessary, the type of ventilation can be changed by changing the set temperatures. Closed cycles have been created in the mode of evaporative cooling, from which the system can exit when necessary, observing the measures to prevent the development of bacteria and molds on the evaporative cooling pads.

Dependencies of the felt temperature on actual temperature, wind speed, relative humidity and age of broilers are derived. This allows a direct calculation of felt temperature in tunnel ventilation. Thus, the system can autonomously maintain conditions that ensure the comfort zone of the broilers in this mode.

The proposed cyber-physical system can only be used in farms that are equipped with all the sensors and microclimate control systems described above. If some of them are missing, they must be installed.

For each individual farm, calculations must be made in relation to maximum broiler number, the installed fans and the number of blind positions. It is determined by the building design. It must be known at which position of the blinds the ventilation supplies the minimum required volume of fresh air for the maximum number of broilers in Minimum ventilation mode.

Data must be entered into the system for the number of blind positions, the position at which the desired fresh air capacity is reached for Minimum ventilation mode and the respective fan powers corresponding to the blind positions to achieve the desired differential pressure. In order to be able to control the heating, whether it is electric, gas, liquid or solid fuel, there must be an external controller that is connected to and controlled by the system.

## Figures and Tables

**Figure 1 animals-13-03252-f001:**
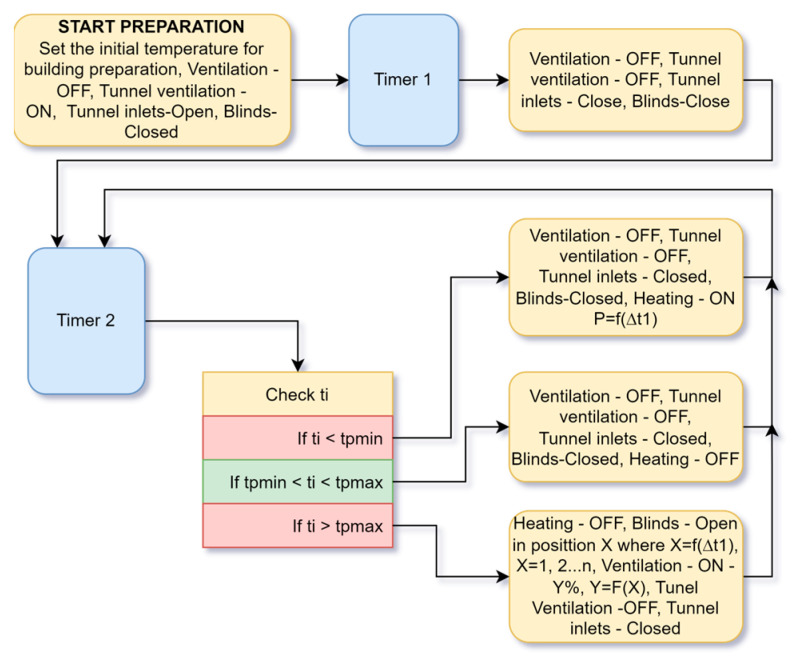
Algorithm to maintain a constant temperature before placing the chickens.

**Figure 2 animals-13-03252-f002:**
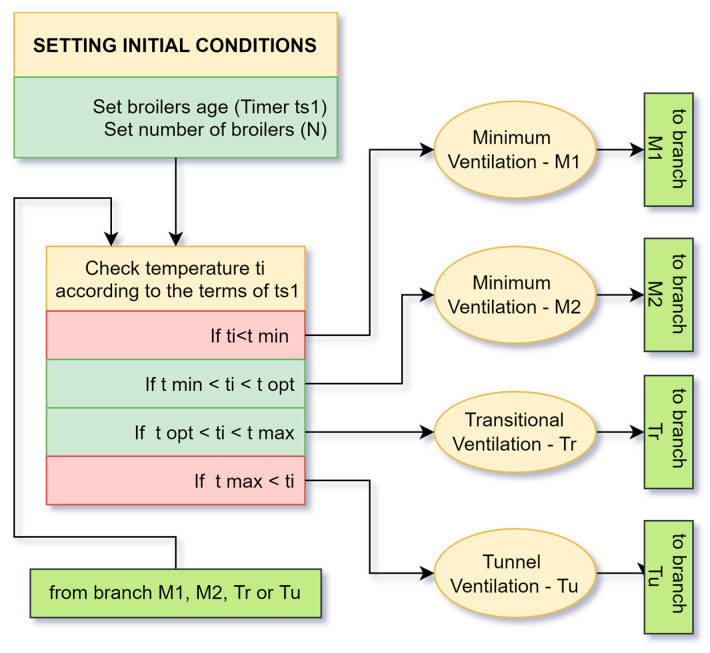
First part of an algorithm for managing a poultry farm.

**Figure 3 animals-13-03252-f003:**
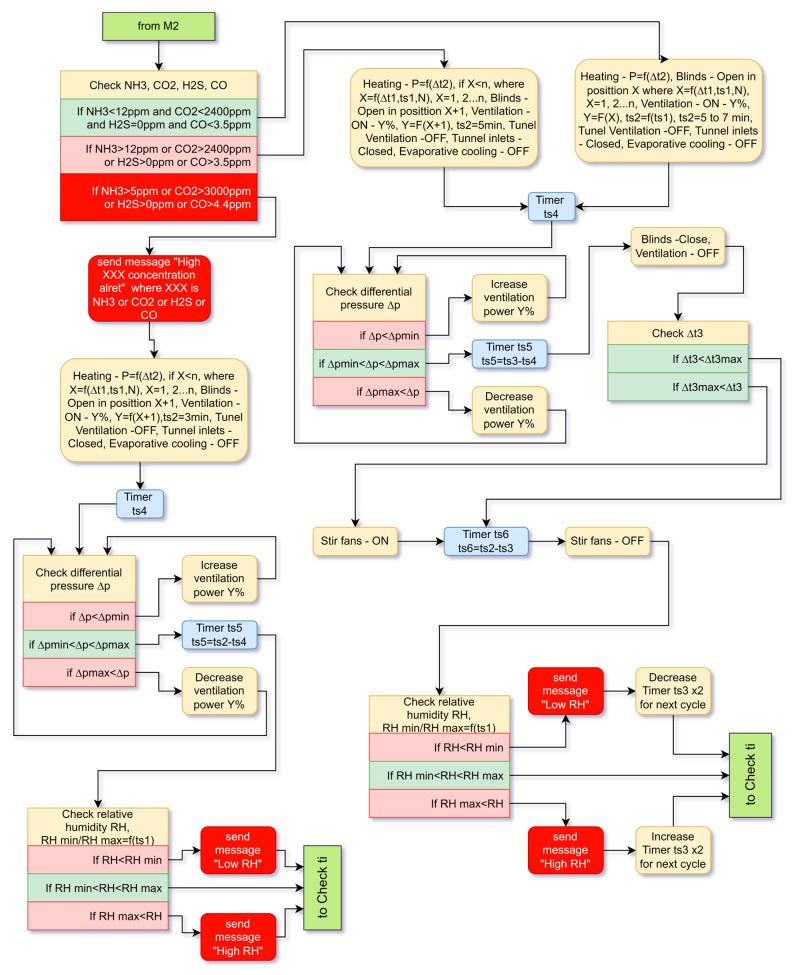
Algorithm for minimum ventilation—branch M2.

**Figure 4 animals-13-03252-f004:**
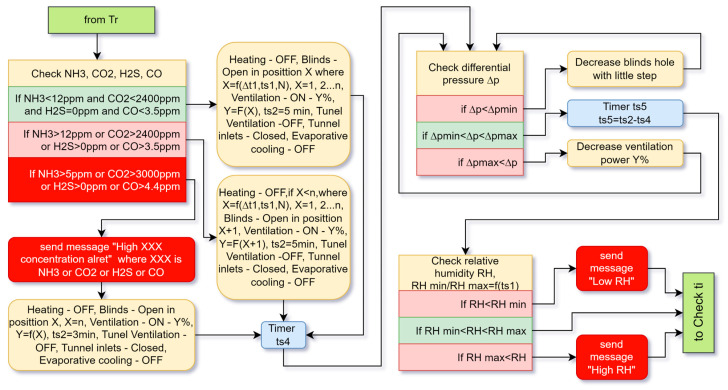
Algorithm for transitional ventilation—branch Tr.

**Figure 5 animals-13-03252-f005:**
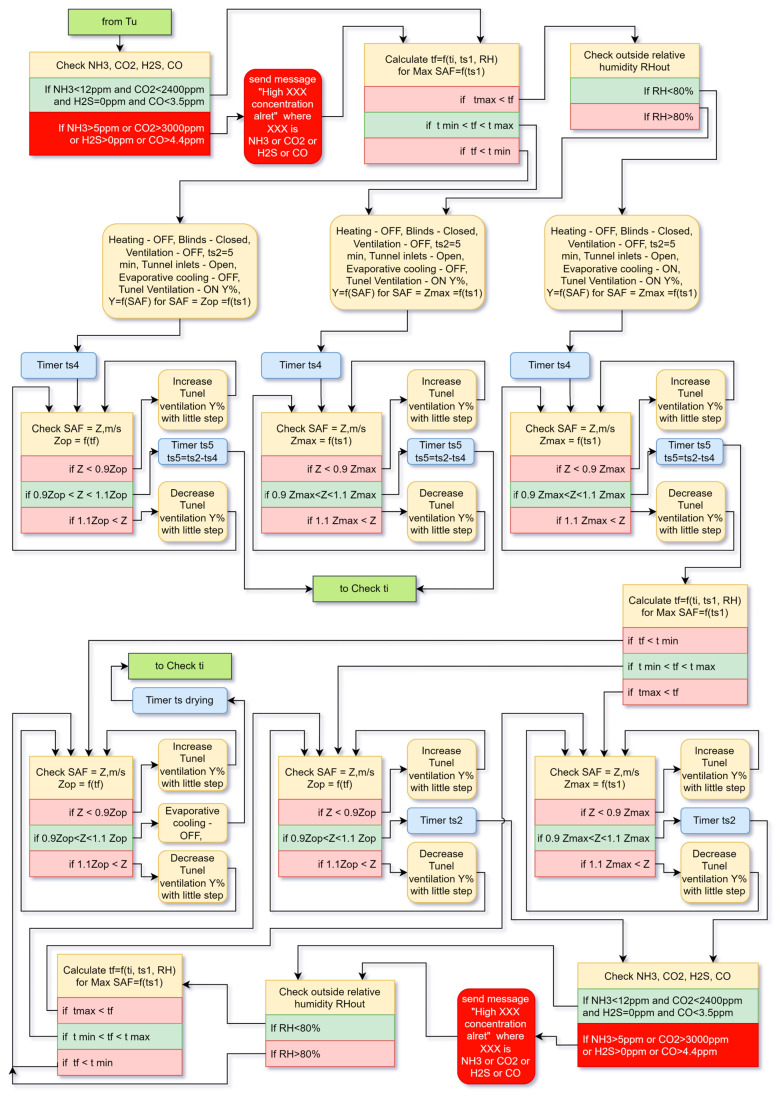
Algorithm for tunnel ventilation—branch Tu.

**Figure 6 animals-13-03252-f006:**
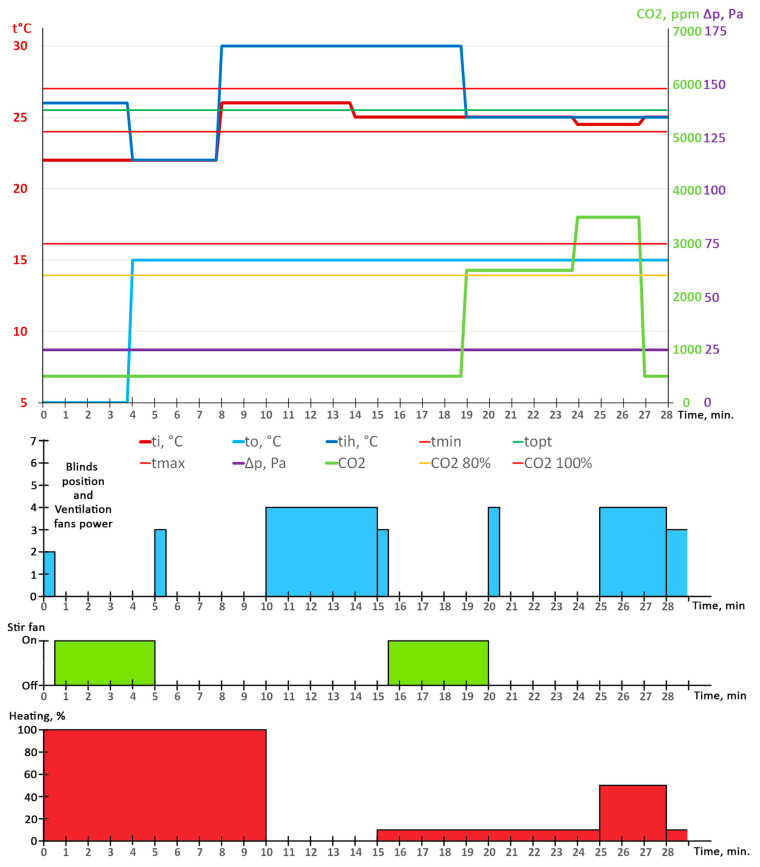
Response of the control systems based on the input data received.

**Figure 7 animals-13-03252-f007:**
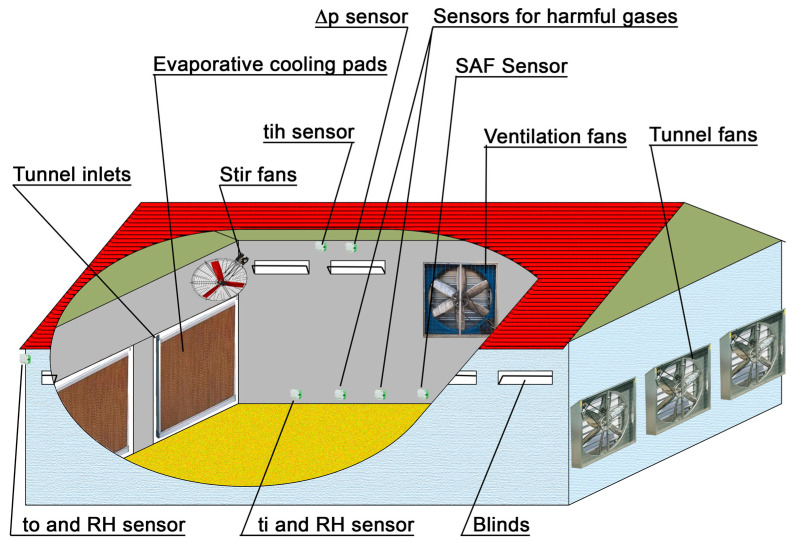
Sensor placement in a poultry farm.

**Table 1 animals-13-03252-t001:** Optimal temperatures and RH for broilers.

№	Broiler Age	Temperature, °C	RH %
		min/max	min/max
1	1–2 days	36/33	72/75
2	3–4 days	33/31	72/75
3	5–6 days	31/30	72/75
4	7–10 days	30/27	70/75
5	11–15 days	27/24	70/75
6	16–20 days	24/22	65/70
7	21–25 days	22/21	65/70
8	26–30 days	21/20	60/65
9	31–35 days	20/19	60/65
10	Over 35 days	18	60/65

**Table 2 animals-13-03252-t002:** Wind chill effect for 7-week-old broilers depending on ambient temperature (°C) and wind speed (m/s).

SAF, m/s	0.5	0.75	1	1.25	1.5	1.75	2	2.25	2.5
t, °C	tcw	tcw	tcw	tcw	tcw	tcw	tcw	tcw	tcw
38	0	0	0	0	0	0	0	0	0
37	0	0	0.1	0.2	0.3	0.45	0.6	0.8	1
36	0	0.1	0.3	0.5	0.8	1.2	1.6	1.9	2.4
35	0	0.15	0.5	0.8	1.2	1.8	2.5	3	3.6
34	0	0.2	0.7	1.2	1.4	2	3	3.6	4.3
33	0	0.35	0.9	1.4	1.6	2.5	3.4	4	4.8
32	0	0.4	1	1.5	1.8	3	3.8	4.7	5.3
31	0	0.45	1.1	1.65	1.95	3.15	3.95	4.9	5.7
30	0	0.5	1.1	1.8	2.15	3.3	4.1	5.1	5.8
29	0	0.54	1.2	1.95	2.3	3.5	4.3	5.3	6
28	0	0.57	1.2	2.1	2.45	3.7	4.5	5.5	6.2
27	0	0.61	1.3	2.2	2.55	3.8	4.6	5.6	6.3
26	0	0.64	1.3	2.3	2.65	3.9	4.7	5.7	6.4
25	0	0.67	1.4	2.4	2.75	4	4.8	5.8	6.5
24	0	0.71	1.5	2.5	2.85	4.1	4.9	5.9	6.6
23	0	0.74	1.6	2.6	2.95	4.2	5	6	x
22	0	0.76	1.7	2.7	3.05	4.3	5.1	x	x
21	0	0.78	1.8	2.75	3.1	4.4	x	x	x
20	0	0.8	1.9	2.8	3.15	x	x	x	x
19	0	0.83	2	2.85	x	x	x	x	x
18	0	0.85	2	x	x	x	x	x	x

**Table 3 animals-13-03252-t003:** Effect of relative humidity RH on felt temperature in broilers.

	∆RH, % ∆RH = RH-RHopt, Rhopt = f(ts1)	1 < ∆RH ≤ 5	5 < ∆RH ≤ 10	10 < ∆RH ≤ 15	15 < ∆RH ≤ 20	20 < ∆RH ≤ 25	25 < ∆RH
ti, °C		tcRH	tcRH	tcRH	tcRH	tcRH	tcRH
35		0.65	1.25	1.80	2.30	2.80	3.20
34		0.60	1.20	1.70	2.25	2.70	3.10
33		0.60	1.15	1.65	2.20	2.60	3.00
32		0.55	1.10	1.55	2.10	2.50	2.90
31		0.55	1.05	1.45	1.95	2.40	2.70
30		0.50	1.00	1.35	1.85	2.30	2.50
29		0.50	0.95	1.30	1.75	2.15	2.30
28		0.45	0.85	1.25	1.65	1.95	2.20
27		0.40	0.75	1.15	1.50	1.80	2.00
26		0.35	0.70	1.05	1.35	1.65	1.90
25		0.30	0.65	0.95	1.20	1.50	1.70
24		0.30	0.60	0.90	1.15	1.40	1.60
23		0.30	0.55	0.80	1.05	1.25	1.45
22		0.30	0.50	0.75	1.00	1.15	1.35
21		0.25	0.45	0.70	0.90	1.05	1.25
20		0.25	0.40	0.60	0.80	0.95	1.10
19		0.20	0.35	0.50	0.65	0.80	0.90
18		0.15	0.25	0.35	0.45	0.55	0.65

**Table 4 animals-13-03252-t004:** Parameter designation.

Designation	Description
ti, °C	Indoor temperature at broiler level
to, °C	Outdoor temperature
tih, °C	Indoor temperature below the roof
tf, °C	Felt temperature
tmin, °C	Minimum comfort zone temperature
tmax, °C	Maximum comfort zone temperature
topt, °C	Optimum growing temperature
tpmin, °C	Minimum room preparation temperature
tpmax, °C	Maximum room preparation temperature
tcw, °C	Temperature correction depending on SAF
RH, %	Indoor relative humidity
RHout, %	Outdoor relative humidity
tcRH, °C	Temperature correction depending on RH
∆t1, °C	∆t1 = ti − to
∆t2, °C	∆t2 = top − ti
∆t3, °C	∆t3 = tih − ti
SAF, m/s	Speed of air flow
∆p, Pa	Differential pressure outdoor/indoor
ts1, day	Age of broilers in days
ts2, min	Ventilation cycle time in minutes
ts3, s	Ventilation operation time for one cycle in s
ts4, s	Time to turn on the fans and open the blinds or tunnel inlets
ts drying	Drying time of the evaporative cooling pads

**Table 5 animals-13-03252-t005:** Simulated data submitted via MQTT broker.

min	ti, °C	to, °C	tih, °C	∆p, Pa	CO_2_
0′	22	5	26	25	500
4′	22	15	22	25	500
8′	26	15	30	25	500
14′	25	15	30	25	500
19′	25	15	25	25	2500
24′	24.5	15	25	25	3500
27′	25	15	25	25	500

## Data Availability

The data presented in this study are available in the article.

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
