# Peer review of "Algorithm for Autonomous Management of a Poultry Farm by a Cyber-Physical System"

_animals, 2023, doi:10.3390/ani13203252_

Round 1
Reviewer 1 Report
This manuscript describes a potentially useful method of environmental control in poultry production systems, that can be controlled by relatively inexpensive sensors, and switching systems, using a simple computer, the Raspberry Pi. It could be of interest to those in the poultry industry, and also to others in intensive animal production industries where good environmental control is important for production or animal welfare.
The Introduction begins with a short but satisfactory section on the importance of the poultry industry, but the main part of the Introduction has two long paragraphs with some unsubstantiated claims, such as the importance of “felt temperature” rather than measured temperature. Some references are given later in part 2, but they belong up here in the Introduction.
Section 2, the specifics of raising broilers gives good coverage of the issues involved, and the need for some form of environmental control.
Section 3, Ventilation, covers the main aspects of maintaining the correct temperature and humidity, but should include a section on harmful gases, CO2, NH3, CO, etc, since in some cases the removal of toxic gases has to take precedence over temperature control.
Section 4, the CPS Algorithm explains how the system makes decisions about which aspects to turn on and off, and the diagrams aid in understanding the flow of information and the decision-making process. There is considerable overlap of decision making in Figure 5 (upper and lower halves of the diagram), and it is not clear whether these must be handled by quite separate duplicated procedures in the program, or whether there are procedures called by the program from different stages of the processing.
Section 5, Test and Result, showed that the system is not actually operating in a poultry production system at this stage, but appeared to confirm that it works in principle, and could be functional in a real production system. The description of events at different times, was hard to follow. Instead of table 5, a figure showing the timeline would be useful, with the inputs, temperature and humidity each minute, and the outputs, fan speed, blind positions, etc shown as they change in response to critical input values. It is not quite clear how many different simulations have been carried out, covering extreme events, to be sure that the system would work as required in all cases.
Section 6. Conclusions, seemed a bit weak. It is not clear whether the proposed system would apply to the majority of broiler production systems, or would need modification for every system, due to differences in the method of heating, cooling and ventilation. If so, there are presumably separate sections of the control program that can be adjusted without changing the main decision-making sections. This might require moving sections about fan speed and blind position to a separate block, but this does not appear to be the case in the flow diagrams.
Some mention could be made of the benefits of environmental control for animal welfare, not just for optimal production.
The writing is good, but there are occasional odd phrases used. Checking by a reader proficient in English, but not associated with the project would be useful.
Author Response
"Please see the attachment."

Reviewer 2 Report
The study presents a Cyber-Physical System (CPS) designed for autonomous management of poultry farms, focusing on optimizing the microclimate for broiler production. By leveraging cost-effective components, such as the Raspberry Pi 4, and the open-source software OpenHAB, the system minimizes human errors, boosts productivity, and reduces waste, ensuring ideal conditions for broiler growth. However, several issues must be addressed before publication.
Paper title: Ensure "CPS" is spelled out if it's not a common abbreviation in open software files.
Lines 34-35: Ensure the logic flows correctly from increased meat consumption to the need for high-level automatic farms.
Abbreviations: Ensure each abbreviation is introduced in its full form before using it in the paper.
Line 43: Add the "Bist, Ramesh Bahadur, et al. "Ammonia emissions, impacts, and mitigation strategies for poultry production: A critical review." Journal of Environmental Management 328 (2023)" citation for the article after mentioning NH3.
Lines 55-57: Insert references for Raspberry Pi 4.
Line 231: Ensure there are notes explaining each variable in the equations.
Lines 322-323: Ensure the language used describes the end of the cycle in scientific terms, rather than in the form of computer code.
Lines 431-434: Recommend the inclusion of images to show the application of CPS.
Author Response
"Please see the attachment."

Reviewer 3 Report
Authors provided robust scientific improvements based on their methodology and results. There are aspects of the manuscript that need expansion and others that need including.
On page 2 lines 42-70, the authors provide the science's background but do not include any citations to the science mentioned in this section of the Introduction. These inclusions need empirical support - scholarly citations for these identified scientific and procedural aspects.
The authors mention "qualified personnel" in the Conclusion (page 16 line 438). This is the beginning of a strong aspect - the connection of science to practice - but the authors only mention this once without details and the "practice" aspect of the scholarship is needed in the Introduction and detailed in the Conclusion respective to how the authors science improves practice with specificity.
Here are some contemporary publications that convey "how" we can leverage our science to improve practitioners (end users; farmers, actors, stakeholders, etc.) by motivating farmer adoption of autonomous poultry system innovations:
https://doi.org/10.1016/j.njas.2019.100315
https://doi.org/10.1007/s13593-016-0380-z
https://doi.org/10.1080/17565529.2017.1411240
https://doi.org/10.3390/s22186833
Author Response
"Please see the attachment."

Round 2
Reviewer 2 Report
After carefully reviewing all the content, since the author has addressed all the comments, I don't have any further comments.
Reviewer 3 Report
The authors sufficiently addressed my concerns warranting expansion and scientific and practice rigor.